# Unexpected Pleiotropic Effects of SGLT2 Inhibitors: Pearls and Pitfalls of This Novel Antidiabetic Class

**DOI:** 10.3390/ijms22063062

**Published:** 2021-03-17

**Authors:** Hideaki Kaneto, Atsushi Obata, Tomohiko Kimura, Masashi Shimoda, Tomoe Kinoshita, Taka-aki Matsuoka, Kohei Kaku

**Affiliations:** 1Department of Diabetes, Endocrinology and Metabolism, Kawasaki Medical School, 577 Matsushima, Kurashiki 701-0192, Japan; obata-tky@med.kawasaki-m.ac.jp (A.O.); tomohiko@med.kawasaki-m.ac.jp (T.K.); masashi-s@med.kawasaki-m.ac.jp (M.S.); kinoshita@med.kawasaki-m.ac.jp (T.K.); 2The First Department of Internal Medicine, Wakayama Medical University, Wakayama 641-8510, Japan; matsuoka@wakayama-med.ac.jp; 3Department of General Internal Medicine 1, Kawasaki Medical School, 577 Matsushima, Kurashiki 701-0192, Japan; kka@med.kawasaki-m.ac.jp

**Keywords:** pancreatic β-cells, insulin resistance, fatty liver, cardio-protection, renal protection

## Abstract

Sodium-glucose co-transporter 2 (SGLT2) inhibitors facilitate urine glucose excretion by reducing glucose reabsorption, leading to ameliorate glycemic control. While the main characteristics of type 2 diabetes mellitus are insufficient insulin secretion and insulin resistance, SGLT2 inhibitors have some favorable effects on pancreatic β-cell function and insulin sensitivity. SGLT2 inhibitors ameliorate fatty liver and reduce visceral fat mass. Furthermore, it has been noted that SGLT2 inhibitors have cardio-protective and renal protective effects in addition to their glucose-lowering effect. In addition, several kinds of SGLT2 inhibitors are used in patients with type 1 diabetes mellitus as an adjuvant therapy to insulin. Taken together, SGLT2 inhibitors have amazing multifaceted effects that are far beyond prediction like some emerging magical medicine. Thereby, SGLT2 inhibitors are very promising as relatively new anti-diabetic drugs and are being paid attention in various aspects. It is noted, however, that SGLT2 inhibitors have several side effects such as urinary tract infection or genital infection. In addition, we should bear in mind the possibility of diabetic ketoacidosis, especially when we use SGLT2 inhibitors in patients with poor insulin secretory capacity.

## 1. Introduction

The number of patients with type 2 diabetes mellitus (T2DM) has markedly increased all over the world due to various changes in lifestyle. The main characteristics of T2DM are insufficient β-cell function and insulin resistance [1,2,3]. Since sodium-glucose cotransporter 2 (SGLT2) contributes to most of renal glucose reabsorption, inhibition of SGLT2 is promising for T2DM [4,5,6,7,8,9,10,11]. In this review article, we show that SGLT2 inhibitors have surprisingly many multifaceted effects beyond prediction like some emerging magical medicine. SGLT2 inhibitors protect pancreatic β-cells from glucose toxicity and mitigate insulin resistance, which finally ameliorates glycemic control. Moreover, while nonalcoholic fatty liver disease (NAFLD) is often brought out in patients with T2DM, it has been shown that SGLT2 inhibitors ameliorate NAFLD. Furthermore, attention has been paid to the fact that SGLT2 inhibitors have cardio-protective and renal protective effects beyond their glucose-lowering effect.

## 2. Concept of Pancreatic β-Cell Glucose Toxicity Found in Type 2 Diabetes Mellitus

The main characteristics of T2DM are insufficient insulin secretion and insulin resistance. Under healthy conditions, blood glucose levels are well controlled by insulin. Under diabetic conditions, however, increased insulin resistance impairs insulin biosynthesis and secretion. First, insulin resistance is brought out by overeating and/or lack of exercise, but enough amount of insulin is secreted so that insulin resistance is compensated. However, when β-cells are exposed to high glucose concentration for a long period of time, β-cell function gradually declines due to overwork. Insulin biosynthesis and secretion decline together with reduction of insulin gene transcription factors MafA and PDX-1 [12,13,14,15,16] (Figure 1). Finally, β-cell number is decreased apoptosis and/or de-differentiation. Such phenomena are called β-cell glucose toxicity. It has been shown, however, that β-cell function is recovered by an appropriate intervention with various anti-diabetes agents [17,18,19,20,21,22]. Chronic hyperglycemia also impairs insulin signaling in insulin target tissues, which leads to the progression of insulin resistance. Therefore, glucose toxicity is involved in insulin resistance in addition to β-cell dysfunction.

MafA is a potent insulin gene transcription factor [12,13,14,15]. MafA expression level is markedly decreased under diabetic conditions but it is preserved after reduction of glucose toxicity with insulin or other anti-diabetic agents. Furthermore, we reported that serum insulin levels were increased and blood glucose levels were decreased by preservation of MafA expression in β-cells in obese type 2 diabetic db/db mice using the Cre-loxP system. In addition, β-cell mass was preserved by MafA overexpression [15]. We think that these findings show that reduction of MafA expression leads to β-cell dysfunction found in T2DM. PDX-1 is also an insulin gene transcription factor and plays an important role in β-cells, but its expression level declines after exposure to chronic hyperglycemia [12,13]. We think that such reduction of PDX-1 also leads to β-cell dysfunction found in T2DM. Indeed, glucose-stimulated insulin secretion was enhanced, and glycemic control was ameliorated by preservation of PDX-1 expression in β-cells using the Cre-loxP system [16]. Taken together, we believe that that reduction of MafA and PDX-1 expression leads to β-cell dysfunction found in T2DM.

Furthermore, while there are several reports indicating the role of insulin signaling in endothelial cells [23,24,25,26], it seems that endothelial dysfunction is also associated with β-cell dysfunction. We assume that since pancreatic islets are easily damaged by ischemia or various stress, β-cell dysfunction is brought out by endothelial dysfunction. Actually, we recently reported that in vascular endothelial cell-specific knockout mice of PDK1, one of the important molecules in insulin signaling, β-cell dysfunction, was brought out together with reduction of vascularity in islets, fall into ischemic state, and increase of inflammation and endoplasmic reticulum stress in β-cells [26]. Since such phenotypes are very similar to the phenomena found under diabetic conditions, we assume that endothelial dysfunction is also, at least in part, associated with β-cell dysfunction found in T2DM.

## 3. SGLT2 Inhibitors Ameliorate Hyperglycemia by Increasing Urinary Glucose Excretion

SGLT2 inhibitors are often used in patients with T2DM worldwide [4,5,6,7,8,9,10,11]. SGLT2 inhibitors lead to amelioration of glycemic control by enhancing urinary glucose excretion. In addition, it is known that, under diabetic conditions, SGLT2 expression in the kidney is augmented, leading to increase renal glucose reabsorption. There is a large amount of evidence worldwide regarding the efficacy and safety of SGLT2 inhibitors in patients with T2DM. Tofogliflozin has high selectivity for SGLT2 and canagliflozin has low selectivity [27], but it is still controversial whether such difference is important or not for our bodies. In addition, dual SGLT1/2 inhibitors were also developed, and their effects are also promising. It was reported that dual SGLT1/2 inhibitors increased GLP-1 levels and reduced postprandial glucose levels [28].

## 4. SGLT2 Inhibitors Have Some Favorable Effects on Pancreatic β-Cells

It has been shown that SGLT2 inhibitors protect pancreatic β-cells from glucose toxicity [29,30,31,32,33,34,35,36]. Indeed, it was reported that an SGLT2 inhibitor luseogliflozin preserved β-cell function in obese type 2 diabetic db/db mice [35]. Pancreatic β-cell mass was significantly increased by luseogliflozin treatment. In addition, β-cell proliferation was increased, and β-cell apoptosis was decreased by such treatment. Insulin biosynthesis and expression levels of MafA and PDX-1 were also enhanced by such treatment [35]. Taken together, SGLT2 inhibitors have some favorable effects for the preservation of β-cell function (Figure 2).

In addition, to focus on the direct effects of SGLT2 inhibitor on β-cells, researchers performed similar experiments for only 1 week with empagliflozin [34]. As a result, lipid metabolism was not influenced, but empagliflozin showed protective effects on β-cells. Insulin, MafA, and PDX-1 expression was increased by empagliflozin treatment. These data indicate that SGLT2 inhibitor directly protect β-cells from glucose toxicity [34] (Figure 2).

Furthermore, we recently compared the protective effects of luseogliflozin on β-cell function and mass between at an early and at advanced stage of diabetes and between after short-term usage and long-term usage using diabetic db/db mice [36]. In terms of the results, β-cell function was preserved by luseogliflozin treatment at an early stage of diabetes, but not at advanced stage. Moreover, β-cell mass was increased by such treatment only at an early stage. Furthermore, when db/db mice were treated with luseogliflozin for a long period of time from an early stage, insulin biosynthesis and secretion were preserved, even at an advanced stage [36]. Taken together, SGLT2 inhibitors have more favorable effects at an early stage, and long-term usage of SGLT2 inhibitors have more favorable effects on β-cells rather than short-term usage (Figure 2).

While various studies were actively performed regarding the role of α-cells [37], it was shown that SGLT2 was present in α-cells and that dapagliflozin enhanced glucagon secretion, leading to the increase of hepatic gluconeogenesis [38]. In spite of such augmentation, blood glucose levels were decreased by dapagliflozin, probably due to the increase of glycosuria. It was also shown that dapagliflozin facilitated glucagon secretion and gluconeogenesis in healthy mice, which limited the reduction of blood glucose levels induced by fasting [38]. It is noted, however, that the direct effect of SGLT2 inhibition on glucagon secretion is currently a controversial topic. For example, one group showed SGLT2 expression in human islets and thought dapagliflozin directly stimulated glucagon secretion through SGLT2 inhibition in α-cells [38], whereas another group showed that SGLT2 expression was not detectable in human or mouse islets [39]. Furthermore, very recently, the importance of SGLT2 activity on glucagon secretion was evaluated by using isolated perfused rat pancreas. As the results, glucagon secretion was decreased by high glucose, but it was not affected by SGLT2 inhibitors. They concluded that increased glucagon secretion by SGLT2 inhibitor was not due to direct effect of SGLT2 on α-cells [40].

## 5. SGLT2 Inhibitors Have Some Favorable Effects on Insulin Resistance

It is known that SGLT2 inhibitors show favorable effects on insulin target tissues. Usage of SGLT2 inhibitors lead to amelioration of fatty liver, reduction of visceral fat mass, and mitigation of insulin resistance [41,42,43,44,45,46,47] (Figure 2). Indeed, tofogliflozin decreased insulin resistance and ameliorated glycemic control in diabetic mice [46]. In this study, C57BL/6 mice were given normal chow or high-fat chow containing tofogliflozin. As a result, tofogliflozin decreased body weight and blood glucose levels in diabetic mice. In addition, adipocyte size and fat mass were reduced by tofogliflozin, which finally enhanced insulin sensitivity [46]. Excess insulin secretion was decreased by tofogliflozin, which led to increase of lipolysis, β-oxidation, and ketone bodies. Such reduction of excess insulin secretion increased gluconeogenesis and decreased lipogenesis. The decrease of lipogenesis led to the decrease of triglyceride content in the liver. Furthermore, hyper-insulinemic euglycemic clamp study showed that insulin resistance was reduced by tofogliflozin mainly due to increase of glucose uptake into skeletal muscle [46]. Taken together, tofogliflozin mitigates insulin resistance mainly through the increase of lipolysis in adipose tissues and glucose uptake into skeletal muscle (Figure 2). In addition, it was shown that ipragliflozin decreased fat mass mainly through the increase of fatty acid oxidation in high-fat diet-induced obese rats and that tofogliflozin reduced fat accumulation in diet-induced obese rats and KKAy mice [41,42].

It has been reported that SGLT2 inhibitors show some favorable effects in patients with T2DM as well [43,44,45]. It was shown that dapagliflozin reduced blood glucose levels and fat mass for a long time in patients with T2DM. In addition, it was shown that dapagliflozin enhanced insulin sensitivity in skeletal muscle but increased hepatic glucose production (HGP) in patients with T2DM. In this study, patients with T2DM were randomized to take dapagliflozin or placebo, and HGP was evaluated using the clamp technique. As a result, HGP as well as plasma glucagon level were increased by dapagliflozin. They showed that the improvement of glycemic control with dapagliflozin enhanced insulin sensitivity in skeletal muscle but increased HGP. It was also reported that HGP and glucagon level were enhanced by empagliflozin in patients with T2DM. We assume that the increase of HGP does not necessarily indicate the aggravation of insulin resistance in the liver. It is possible that such enhancement of HGP with SGLT2 treatment is, at least in part, related to the decreased ratio of serum insulin/glucagon.

## 6. SGLT2 Inhibitors Have Some Favorable Effects on Nonalcoholic Fatty Liver Disease (NAFLD) Though the Alleviation of Hyperinsulinemia

NAFLD is often observed in clinical practice, and the number of patients with NAFLD has been markedly increased all over the world. NAFLD includes a simple fatty liver and nonalcoholic steatohepatitis (NASH), together with fibrosis and inflammation [48,49,50,51]. Patients with T2DM have higher risk of NAFLD, and the presence of T2DM tends to aggravate NAFLD [52,53,54]. It was also reported that patients with T2DM had higher risk for liver cirrhosis and hepatocellular carcinoma [55]. Previously, hepatitis C virus and heavy usage of alcohol were the main two causes of liver cirrhosis and hepatocellular carcinoma. However, recently, hepatitis C has become curable and it has been noted that various liver disorders are brought out in subjects without drinking alcohol. Diabetes mellitus has become one of the risk factors of liver disorders instead of hepatitis C virus and heavy usage of alcohol, and recently malignancies including hepatocellular carcinoma have sometimes been regarded as diabetic complications in addition to classical diabetic micro- and macro-complications. It is well known that thiazolidinediones have favorable effects on NAFLD [56,57]. However, thiazolidinediones often increase body weight, and thereby usage of thiazolidinediones is not necessarily appropriate in patients with T2DM and NAFLD.

It has been noted that SGLT2 inhibitors have some favorable effects on the liver [58,59,60,61]. It was reported that ipragliflozin and pioglitazone were effective on NAFLD in patients with T2DM and NAFLD [58]. The patients were randomized to take ipragliflozin or pioglitazone. As a result, the liver-to-spleen ratio on computed tomography (CT) was increased in either ipragliflozin or pioglitazone group. Diabetes markers such as HbA1c and plasma glucose levels were similarly reduced in the two groups. However, body weight and visceral fat area were significantly reduced only in the ipragliflozin group. Moreover, they showed that both drugs had favorable effects on NAFLD and glycemic control to a similar extent in patients with T2DM and NAFLD [58]. We also reported the effects of three oral anti-diabetic drugs (dapagliflozin, pioglitazone, and glimepiride) on NAFLD in a prospective clinical trial in patients with T2DM and NAFLD [60]. We evaluated the alteration of the liver-to-spleen ratio on CT. As a result, glycemic control was improved in the three groups to almost the same extent. Significant reduction of body weight and visceral fat area was observed in the dapagliflozin group, and marked increase of serum adiponectin level was observed in the pioglitazone group. The liver-to-spleen ratio was significantly increased by dapagliflozin or pioglitazone, but not by glimepiride. However, these two drugs seemed to have different mechanism of action. Dapagliflozin showed favorable effects on NAFLD by alleviating hyperinsulinemia, while pioglitazone exerted such effects by increasing adiponectin levels [60]. Taken together, SGLT2 inhibitors are likely to have some favorable effects on NAFLD by mitigating hyperinsulinemia.

## 7. SGLT2 Inhibitors Have Substantial Cardio-Protective Effects

Various kinds of SGLT2 inhibitors have been used worldwide. There are various clinical trials about cardio-protective effects of SGLT2 inhibitors [62,63,64,65,66,67,68]. For example, the effects of empagliflozin on cardiovascular morbidity and mortality were shown in patients with T2DM and high cardiovascular risk. Patients were randomized to take empagliflozin or placebo. The primary composite cardiovascular events were observed in 10.5% patients taking empagliflozin and in 12.1% patients taking placebo. Hazard ratio (HR) in empagliflozin group was 0.86. Furthermore, there were significant differences in rates of death from cardiovascular causes between empagliflozin and placebo group (3.7% vs. 5.9%), hospitalization for heart failure (2.7% vs. 4.1%), and death from any cause (5.7% vs. 8.3%). Taken together, patients with T2DM taking empagliflozin showed the lower rate of primary composite cardiovascular outcome and lower rate of all-cause death compared to those taking placebo. In contrast, myocardial infarction and stroke were not well decreased.

The CANVAS program is another large-scale clinical trial examining the effects of SGLT2 inhibitors on cardiovascular events. The CANVAS program collected the data from two trials involving patients with T2DM and high cardiovascular risk. Patients in both trials were randomized to take canagliflozin or placebo. The rate of primary composite cardiovascular events was lower in patients taking canagliflozin compared to placebo (HR, 0.86). Taken together, it is likely that canagliflozin reduces the risk of cardiovascular events in patients with T2DM and an elevated risk of cardiovascular disease.

While SGLT2 inhibitors decreased the combined risk of cardiovascular death and hospitalization for heart failure in patients with or without T2DM in dapagliflozin and prevention of adverse-outcomes in heart failure DAPA-HF trial and EMPEROR-Reduced trial, meta-analyses of the two trials were performed recently in which the primary endpoint was time to all-cause death. As a result, the estimated treatment effect was a 13% reduction in all-cause death (HR, 0.87) and 14% reduction in cardiovascular death (HR, 0.86). The usage of SGLT2 inhibitors reduced the combined risk of cardiovascular death by 26% (HR, 0.74).

It is known that heart failure dominantly influences the prognosis in patients with T2DM. Heart failure with low ejection fraction (EF) can be treated with diuretics and/or anti-hypertensive drugs, but there are no promising drugs for heart failure with preserved ejection fraction, which is characterized by left ventricular diastolic dysfunction. It has been shown recently that SGLT2 inhibitors have some favorable effects on heart failure with preserved EF. The reduction in inflammatory cytokine signaling by SGLT2 inhibitors may explain such molecular mechanism. It is known that SGLT2 inhibitors decrease various cardiovascular risks by improving glucose and lipid metabolism. Moreover, it is well known that hypoglycemia increases cardiovascular events, and we should avoid hypoglycemia especially in patients with cardiovascular event history. The risk of hypoglycemia with SGLT2 inhibitors is quite low, especially in patients with its monotherapy.

It has been thought that increase of ketone bodies are not favorable for our body. For example, it is well known that increased ketone bodies due to lack of insulin brings out ketoacidosis. However, it has been shown recently that some amounts of ketone bodies have favorable effects on the heart. SGLT2 inhibitors usually reduce a ratio of insulin to glucagon. When this ratio is decreased, acetyl-CoA is converted to acetoacetate and β-hydroxybutyrate in the liver. Such β-hydroxybutyrate is carried to the heart in the bloodstream and taken up into the heart. β-Hydroxybutyrate is converted to acetoacetate in the heart, which leads to increased acetyl-CoA. Increased acetyl-CoA activates the TCA cycle and electron transporter chain in mitochondria and eventually increases ATP production in the heart [64] (Figure 3). Furthermore, increase of hematocrit by SGLT2 inhibitors is likely to facilitate ATP production by increasing oxygen supply.

## 8. SGLT2 Inhibitors Have Substantial Renal Protective Effects

While SGLT2 inhibitors have been used worldwide, there are various large-scale clinical trials about renal protective effects of them [63,64,65,69,70,71,72,73,74,75]. It was reported that canagliflozin had favorable effects on renal outcomes. The CANVAS program collected data from two trials with patients with T2DM and high cardiovascular risk. Patients in each trial were randomized to take canagliflozin or placebo [65]. As the results, canagliflozin treatment showed favorable effects on the progression of albuminuria (HR, 0.73). Furthermore, canagliflozin treatment showed favorable effects on the primary composite renal outcome (HR, 0.60) [65].

CREDENCE trial is another large-scale clinical study examining the effects of SGLT2 inhibitors on renal function. In this trial, patients with T2DM and albuminuric chronic kidney disease were randomized to take canagliflozin or placebo. eGFR of each patient was 30 to <90 mL/min/1.73m^2^, and all patients took renin–angiotensin system inhibitors. As a result, canagliflozin treatment reduced the primary composite renal outcome by 30% (HR, 0.70). Moreover, such treatment reduced the risk of end-stage kidney disease, doubling of serum creatinine, or renal-cause death by 34% (HR, 0.66), and reduced the risk of end-stage kidney disease by 32% (HR, 0.68) [69]. Taken together, SGLT2 inhibitor canagliflozin decreased the risk of renal failure in patients with T2DM and kidney disease for a relatively short period of time.

We also reported that effects of canagliflozin on albuminuria in patients with T2DM. The alteration of urinary albumin excretion was positively correlated with the alteration of systolic blood pressure. Furthermore, the alteration of blood pressure was an independent factor for the alteration of urinary albumin excretion in multivariate analysis; the lower the blood pressure, the better the albuminuria [76]. Taken together, canagliflozin reduces urinary albumin excretion by reducing blood pressure in patients with T2DM (Figure 4).

It has been clearly shown recently that SGLT2 inhibitors have renal protective effects. SGLT2 inhibitors improve renal outcomes by reducing urinary albumin excretion. There are various possible mechanisms for renal protective effects of SGLT2 inhibitors including the reduction of blood pressure, increase of ketone body production, increase of sirtuin-1 expression, and constriction of afferent arteriole through the tubule-glomerular feedback system (Figure 4).

First, it was reported that the single-nephron glomerular filtration rate in diabetic mice was higher compared to its control, but it was lower after treatment with empagliflozin. In vivo imaging also revealed concomitant afferent arteriolar dilation and increased glomerular permeability of albumin in diabetic mice, which was ameliorated after the treatment with empagliflozin [77]. Furthermore, they showed that empagliflozin increased urinary adenosine excretion, which led to reduction of hyperfiltration through afferent arteriolar constriction [77]. Second, it was reported that ketone bodies were increased by empagliflozin, which led to preservation of ATP level in the kidney. Furthermore, they showed ketone bodies reduced proteinuria in diabetic mice [78]. Third, while it was known that sirtuin-1 played important roles in various tissues as an anti-aging factor by reducing oxidative stress and inflammation, it was reported that under diabetic conditions, situin-1 expression in the kidney was decreased, which led to the progression of diabetic nephropathy [79,80]. It was also shown that canagliflozin increased sirtuin-1 expression in the kidney of diabetic mice and that SIRT1 expression levels were negatively correlated with SGLT2 expression levels in renal biopsy specimens from humans [79,80]. Taken together, SGLT2 inhibitors exert renal protective effects through a variety of pathways such as reduction of blood pressure, increase of ketone body production, increase of sirtuin-1 expression, and constriction of afferent arteriole through the tubule-glomerular feedback system (Figure 4).

## 9. SGLT2 Inhibitors Are Useful as an Adjuvant Therapy to Insulin Preparation in Patients with Type 1 Diabetes Mellitus

There have been various clinical trials using SGLT2 inhibitors in patients with type 1 diabetes mellitus (T1DM). Dapagliflozin was approved as an adjuvant therapy to insulin preparation in patients with T1DM who have poor glycemic control [81,82]. In Japan, ipragliflozin as well as dapagliflozin can be used as an adjuvant therapy to insulin in patients with T1DM. As an adjuvant therapy to insulin, SGLT2 inhibitors decrease blood glucose levels with smaller dose of total daily insulin preparation. In addition, since SGLT2 inhibitors decrease body weight and insulin preparation increases body weight, starting SGLT2 inhibitor and decreasing dose of insulin preparation leads to reduction of body weight and amelioration of insulin sensitivity. When insulin dose is reduced, the risk of hypoglycemia is also decreased. Since hypoglycemia brings out various clinical problems such as acute coronary syndrome, fundus hemorrhage, and unconscious hypoglycemia due to lack of catecholamine secretion, we should avoid hypoglycemia in clinical practice. In addition, since hypoglycemia damages the brain and facilitates the development of dementia, it is very important to reduce possible hypoglycemia risk, especially in elder patients with T1DM. However, since there is some concern about diabetic ketoacidosis caused by SGLT2 inhibitors, we should bear in mind the possibility of diabetic ketoacidosis, especially in patients with T1DM.

## 10. Side Effects of SGLT2 Inhibitors and Precautions When Using Them

As described above, SGLT2 inhibitors have amazing multifaceted effects that are far beyond prediction. It is noted, however, that SGLT2 inhibitors have several side effects such as urinary tract infection or genital infection. Indeed, it was reported that although the effects of SGLT2 inhibitors on glycemic control were comparable to those observed in large-scale clinical trials, there were safety concerns such as the frequent occurrence of urinary tract or genital infection [83]. In addition, as described above, there is some concern about diabetic ketoacidosis caused by SGLT2 inhibitors, and thus we should bear in mind the possibility of diabetic ketoacidosis, especially in patients with T1DM. Renal sodium reabsorption and glucose metabolism are closely tied with acid–base balance, and many theories have been proposed to explain the increased risk of ketoacidosis (either hyper- or eu-glycemic) in patients on SGLT2 inhibitors. Together with genital and urinary infections (sometimes severe, limiting patient compliance or requiring hospitalization for urosepsis and pyelonephritis), the risk of ketoacidosis associated with SGLT2 inhibitors may generate confusion among treating physicians.

Recently, a new pandemic was induced by coronavirus infectious disease 2019 (COVID-19) worldwide. The mortality in patients with COVID-19 is extremely high, and the main reason of death is severe pneumonia [84]. In addition, the mortality in patients with both COVID-19 and diabetes was found to be very high [85,86]. SGLT2 inhibitors may increase the likelihood of COVID-19-related ketoacidosis in patients with severe insulin deficiency in patients with T1DM or T2DM [87]. Therefore, we should always consider the benefits and the demerits of SGLT2 inhibitors.

## 11. Conclusions

SGLT2 inhibitors decrease blood glucose levels by increasing urinary glucose excretion and show amazing multifaceted effects that are far beyond prediction like some emerging magical medicine. First, SGLT2 inhibitors have some favorable effects on pancreatic β-cell function. Second, SGLT2 inhibitors mitigate insulin resistance in insulin target tissues. Third, SGLT2 inhibitors have some favorable effects on NAFLD, at least in part due to mitigation of hyperinsulinemia. Fourth, SGLT2 inhibitors have cardio-protective effects. Indeed, in several clinical trials, SGLT2 inhibitors showed some favorable effects on cardiovascular morbidity in patients with T2DM. Fifth, SGLT2 inhibitors have renal protective effects. Indeed, in several clinical trials, SGLT2 inhibitors showed some favorable effects on the progression of albuminuria in patients with T2DM. Sixth, SGLT2 inhibitors are useful as an adjuvant therapy to insulin preparation in patients with T1DM. It is noted, however, that SGLT2 inhibitors have several side effects such as urinary tract infection or genital infection. Taken together, we believe that SGLT2 inhibitors have amazing multifaceted effects beyond prediction, and thereby SGLT2 inhibitors would be very promising from the clinical point of view in a variety of aspects, although we have to be careful of their usage in several points.

## Figures and Tables

**Figure 1 ijms-22-03062-f001:**
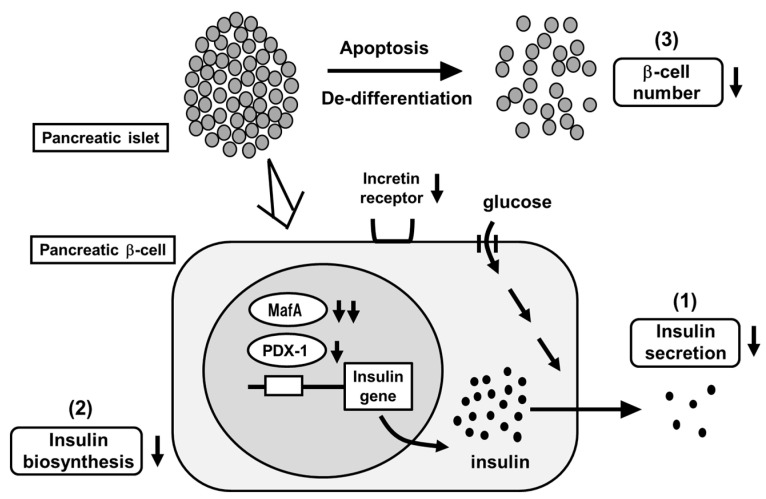
Pancreatic β-cell glucose toxicity found in type 2 diabetes mellitus (T2DM). When pancreatic β-cells are exposed to chronic hyperglycemia, β-cell function gradually declines. First, insulin secretion is reduced. Second, insulin biosynthesis is reduced together with reduction of MafA and PDX-1 expression. Third, β-cell number is decreased through the process of apoptosis and/or de-differentiation.

**Figure 2 ijms-22-03062-f002:**
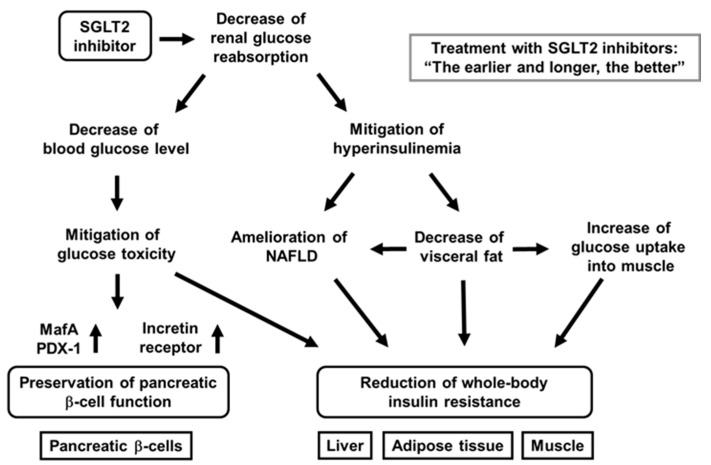
Favorable effects of sodium-glucose cotransporter 2 (SGLT2) inhibitors on pancreatic β-cells and various insulin target tissues. SGLT2 inhibitors ameliorates glycemic control and alleviates hyperinsulinemia by reducing renal glucose reabsorption. Amelioration of glycemic control reduces glucose toxicity, which finally preserves β-cell function and mitigation of insulin resistance in insulin target tissues. Alleviation of hyperinsulinemia ameliorates fatty liver or nonalcoholic fatty liver disease (NAFLD), reduces visceral fat, and increases glucose uptake into skeletal muscle, all of which finally lead to mitigation of whole-body insulin resistance.

**Figure 3 ijms-22-03062-f003:**
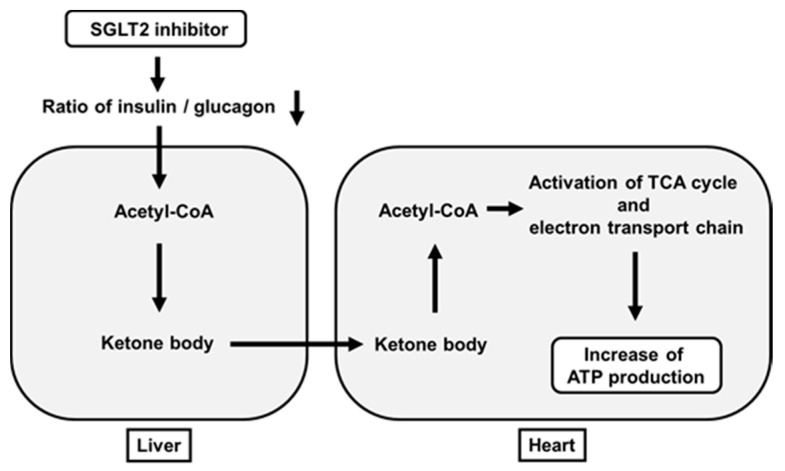
Favorable effects of SGLT2 inhibitors on ATP production in the heart beyond their glucose-lowering effect. SGLT2 inhibitors decrease a ratio of insulin to glucagon, which leads to increased ketone bodies (acetoacetate and β-hydroxybutyrate) in the liver. Such ketone bodies are carried to the heart. After this, they activate the TCA cycle and electron transporter chain in mitochondria and finally increase ATP production in the heart.

**Figure 4 ijms-22-03062-f004:**
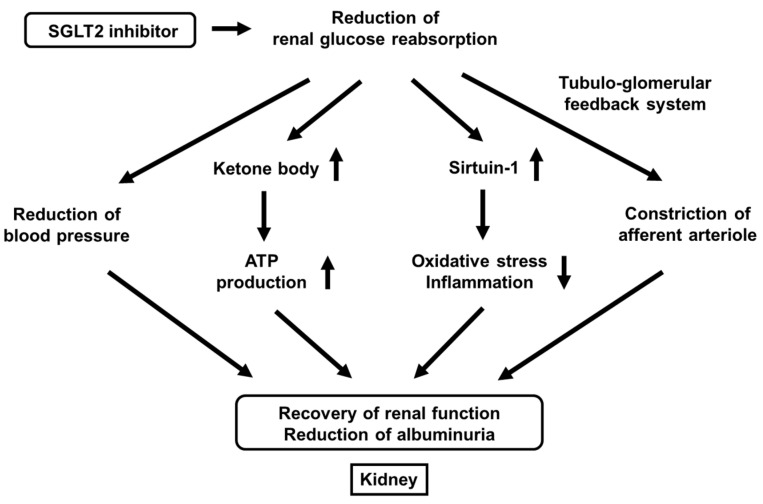
Favorable effects of SGLT2 inhibitors on renal function beyond their glucose-lowering effect. SGLT2 inhibitors reduce renal glucose reabsorption, which leads to the reduction of blood pressure, increase of ketone bodies, increase of sirtuin-1 expression, and constriction of afferent arteriole. All of such alterations lead to the recovery or renal function and the reduction of albuminuria.

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
