# Peer review of "Unexpected Pleiotropic Effects of SGLT2 Inhibitors: Pearls and Pitfalls of This Novel Antidiabetic Class"

_ijms, 2021, doi:10.3390/ijms22063062_

Round 1

Reviewer 1 Report

In this narrative review, Kaneto and colleagues addressed the protective effects of SGLT2 inhibitors against beta cell dysfunction and cardiovascular and renal damage, which have been proven to be - at least to some extent - independent of their glucose-lowering activity. Overall, the review topic is interesting, well-organized and with nicely presented figures. Even though a number of similar reviews can be found in the medical literature, a point of novelty is represented by the integration of clinical and molecular perspectives, especially when detailing the role of SGLT2 inhibitors in the regulation of MafA/PDX1 expression in pancreatic beta cells and preservation of insulin secretion (Section 2). However, several flaws have been found, and I would suggest major changes prior of acceptance for publication.

Major commens:

- Authors are strongly encouraged to provide a more balanced overview of SGLT2 inhibitors, in light of their potential infectious side effects, risk of ketoacidosis decompensation/volume depletion, and other disadvantages, which are being found out often far more frequent than those reported in clinical trials. I would suggest to modify the title accordingly.

-Also, in order to further expand the scientific value of the present work, I would invite the Authors to slightly undertone self-referentiality and consider other groups as suitable references (i.e. J Diabetes Res. 2019;2019:3971060. doi:10.1155/2019/3971060 ; Int J Environ Res Public Health. 2020;17(10):3664. doi:10.3390/ijerph17103664).

-Section 4, lines 161-171: The direct effect of SGLT2 inhibition/dapagliflozin on glucagon secretion in humans is currently a controversial topic. I would suggest discussing and referring to Diabetologia. 2019 Jun;62(6):1011-1023. doi: 10.1007/s00125-019-4849-6 for a more balanced overview.

-Section 9, lines 392-400:  Adding SGLT2 inhibitors to basal/MDI insulin can either reduce the insulin dose or prevent the progressive augmentation over time that is typical of both type 2 and type 1 diabetes, reflecting the progressive rises in body fat content and insulin resistance. It is true that insulin-induced hypoglycemia can be particularly harmful to cardiovascular and neurological health, however, the use of SGLT2 in elderly patients with long-lasting diabetes (either type) and microvascular complications/comorbidities, is a debated issue that should deserve a longer discussion. Elderly patients are inevitably more prone to severe side effects of SGLT2 inhibitors, and weight loss is not necessarily a therapeutic goal in this special population. In view of its clinical importance, I would suggest expanding and reorganizing this section.

Minor comments:

Fig. 2: The box “Therapy with SGLT2 inhibitors…” should be in a different color (i.e. a gray tone). All insulin target tissues names should be singular (i.e. adipose tissue). Please check and correct the beta (β) symbol in figure legends.

Fig. 3 legend: Please rewrite lines 332-333 in something like “Ketone bodies are carried through the bloodstream to the heart where they activate the TCA cycle and the mitochondrial electron transporter chain, thereby increasing ATP production”.

Author Response

Response letter to Reviewer 1’s comments

In this narrative review, Kaneto and colleagues addressed the protective effects of SGLT2 inhibitors against beta cell dysfunction and cardiovascular and renal damage, which have been proven to be - at least to some extent - independent of their glucose-lowering activity. Overall, the review topic is interesting, well-organized and with nicely presented figures. Even though a number of similar reviews can be found in the medical literature, a point of novelty is represented by the integration of clinical and molecular perspectives, especially when detailing the role of SGLT2 inhibitors in the regulation of MafA/PDX1 expression in pancreatic beta cells and preservation of insulin secretion (Section 2). However, several flaws have been found, and I would suggest major changes prior of acceptance for publication.

We appreciate the insights and helpful comments and have revised the manuscript according to your suggestions.

Major commens:

- Authors are strongly encouraged to provide a more balanced overview of SGLT2 inhibitors, in light of their potential infectious side effects, risk of ketoacidosis decompensation/volume depletion, and other disadvantages, which are being found out often far more frequent than those reported in clinical trials. I would suggest to modify the title accordingly.

Thank you very much for you valuable suggestion. According to you kind suggestion, we added the description in abstract (page 3, lines 1-4 from the bottom) and newly added the section 10 entitled “Side effects of SGLT2 inhibitors and precautions when using them” (page 20). In addition, according to your kind suggestion, we changed the title as follows: “SGLT2 inhibitors have amazing pleiotropic effects which is far beyond prediction although we have to be careful in their usage in several points”.

-Also, in order to further expand the scientific value of the present work, I would invite the Authors to slightly undertone self-referentiality and consider other groups as suitable references (i.e. J Diabetes Res. 2019;2019:3971060. doi:10.1155/2019/3971060 ; Int J Environ Res Public Health. 2020;17(10):3664. doi:10.3390/ijerph17103664).

Thank you very much for you valuable suggestion. According to you kind suggestion, we deleted several old references from our laboratory. Instead of it, we added the description by referring several papers from other groups including the references which you kindly mention in the revised version of the manuscript (page 20).

-Section 4, lines 161-171: The direct effect of SGLT2 inhibition/dapagliflozin on glucagon secretion in humans is currently a controversial topic. I would suggest discussing and referring to Diabetologia. 2019 Jun;62(6):1011-1023. doi: 10.1007/s00125-019-4849-6 for a more balanced overview.

Thank you very much for you valuable suggestion. According to you kind suggestion, we amended the description by referring to the paper which you kindly mention in the revised version (page 9, lines 3-16)

-Section 9, lines 392-400:  Adding SGLT2 inhibitors to basal/MDI insulin can either reduce the insulin dose or prevent the progressive augmentation over time that is typical of both type 2 and type 1 diabetes, reflecting the progressive rises in body fat content and insulin resistance. It is true that insulin-induced hypoglycemia can be particularly harmful to cardiovascular and neurological health, however, the use of SGLT2 in elderly patients with long-lasting diabetes (either type) and microvascular complications/comorbidities, is a debated issue that should deserve a longer discussion. Elderly patients are inevitably more prone to severe side effects of SGLT2 inhibitors, and weight loss is not necessarily a therapeutic goal in this special population. In view of its clinical importance, I would suggest expanding and reorganizing this section.

Thank you very much for you valuable suggestion. According to you kind suggestion, as described above, we added the description in abstract (page 3, lines 1-4 from the bottom) and newly added the section 10 entitled “Side effects of SGLT2 inhibitors and precautions when using them” (page 20). In addition, we changed the title as follows: “SGLT2 inhibitors have amazing pleiotropic effects which is far beyond prediction although we have to be careful in their usage in several points”.

Minor comments:

Fig. 2: The box “Therapy with SGLT2 inhibitors…” should be in a different color (i.e. a gray tone). All insulin target tissues names should be singular (i.e. adipose tissue). Please check and correct the beta (β) symbol in figure legends.

Thank you very much for you valuable suggestion. According to you kind suggestion, we changed the box “Treatment with SGLT2 inhibitors ---” into a gray color in Fig. 2. In addition, we made all insulin target names singular in Fig. 2 and corrected the beta symbol in figure legends in the revised version.

Fig. 3 legend: Please rewrite lines 332-333 in something like “Ketone bodies are carried through the bloodstream to the heart where they activate the TCA cycle and the mitochondrial electron transporter chain, thereby increasing ATP production”.

Thank you very much for you valuable suggestion. According to you kind suggestion, we amended the description about this point in the revised version as follows: “Such ketone bodies are carried to the heart. After then, they activate TCA cycle and electron transporter chain in mitochondria and finally increase ATP production in the heart” in Fig. 3 legend in the revised version.

Thank you very much again for your thoughtful comments that have led to strengthening our manuscript.

Reviewer 2 Report

The purpose of this review was to compile recent evidence for the protect effects of SGLT2 inhibitors beyond glucose homeostasis. The review provides a comprehensive account of the role of SGLT2 inhibitors in the regulation of glucose and insulin sensitivity, liver function, renal function, and cardiovascular disease. 

Overall, this was an excellent review.  It was well written, with minor English edits required.  The level of detail in each section was sufficient to provide the reader with adequate information.

The diagrams and figures were excellent.  They helped pull the entire review together. 

One minor suggestion is to number the sequence of events detailed in figure 1.

The citations are a bit old for a review paper.  The majority of sources are 7 to 10 years old.  It is preferred to stick to the most recent 5 years to capture the new findings rather that restating what was likely covered by past reviews.  Indeed, a search through pubmed shows many SGLT2 review articles.  However, the novelty of this paper is the inclusiveness of multiple physiological systems, where as most previous publications focused on specific system effects of SGLT2 inhibitors.  Therefore, the older publications could be justified due to the overarching focus of this review.

Author Response

Response letter to Reviewer 2’s comments

The purpose of this review was to compile recent evidence for the protect effects of SGLT2 inhibitors beyond glucose homeostasis. The review provides a comprehensive account of the role of SGLT2 inhibitors in the regulation of glucose and insulin sensitivity, liver function, renal function, and cardiovascular disease. 

Overall, this was an excellent review.  It was well written, with minor English edits required.  The level of detail in each section was sufficient to provide the reader with adequate information.

The diagrams and figures were excellent.  They helped pull the entire review together. 

We appreciate the insights and helpful comments and have revised the manuscript according to your suggestions.

One minor suggestion is to number the sequence of events detailed in figure 1.

Thank you very much for you valuable suggestion. According to you kind suggestion, we added the number of the sequence of events in Fig. 1 in the revised version.

The citations are a bit old for a review paper.  The majority of sources are 7 to 10 years old.  It is preferred to stick to the most recent 5 years to capture the new findings rather that restating what was likely covered by past reviews.  Indeed, a search through pubmed shows many SGLT2 review articles.  However, the novelty of this paper is the inclusiveness of multiple physiological systems, where as most previous publications focused on specific system effects of SGLT2 inhibitors.  Therefore, the older publications could be justified due to the overarching focus of this review.

Thank you very much for you valuable suggestion. According to you kind suggestion, we deleted old references and newly added new papers as references. We newly added Refs. 3-7, Refs. 9-11, Refs. 20-22, Refs. 29-33, Refs. 39-40, Refs. 53-54, Ref. 61, Refs. 83-87 in the revised version of the manuscript.

Thank you very much again for your thoughtful comments that have led to strengthening our manuscript.

Round 2

Reviewer 1 Report

Dear Authors,

Thank you for your revision. The manuscript has now been significantly improved and most of the comments raised by this reviewer have been resolved. However, I would like to consider minor changes prior of acceptance:

Title: The pdf version of the revised manuscript does not show any substantial change in the review title. I would suggest something like “Unexpected pleiotropic effects of SGLT2 inhibitors: pearls and pitfalls of this novel antidiabetic class” as appropriate.

Section 10: Even though the purpose of this review is not to assess all safety concerns associated with SGLT2 inhibitors use and clinical tips, I would recommend to further expand this section for a more balanced overview of this drug class’ effects. I.e., renal sodium reabsorption and glucose metabolism are closely tied with acid-base balance, and many theories have been proposed to explain the increased risk of ketoacidosis (either hyper- or eu-glycemic) in patients on SGLT2 inhibitors. Together with genital and urinary infections (sometimes severe, limiting patient compliance or requiring hospitalization for urosepsis and pyelonephritis), the risk of ketoacidosis associated with SGLT2 inhibitors use may generate confusion among threating physicians.

Finally, I would recommend checking references order and numbers throughout the manuscript text.

Author Response

Thank you very much for your valuable suggestion.  

According to your kind suggestion, we changed the title as follows: "Unexpected pleiotropic effects of SGLT2 inhibitors: pearls and pitfalls of this novel antidiabetic class"

In section 10, we added the description which you kindly let us know in the revised version (page18, lines 9-14).

Finally, we carefully checked the reference numbers once more, and amended several points (page 12, line line 14; page 19, line 5).

Thank you very much again for your valuable suggestion.